# Dexterous Functional Grasping

**Ananye Agarwal   Shagun Uppal   Kenneth Shaw   Deepak Pathak**
Carnegie Mellon University

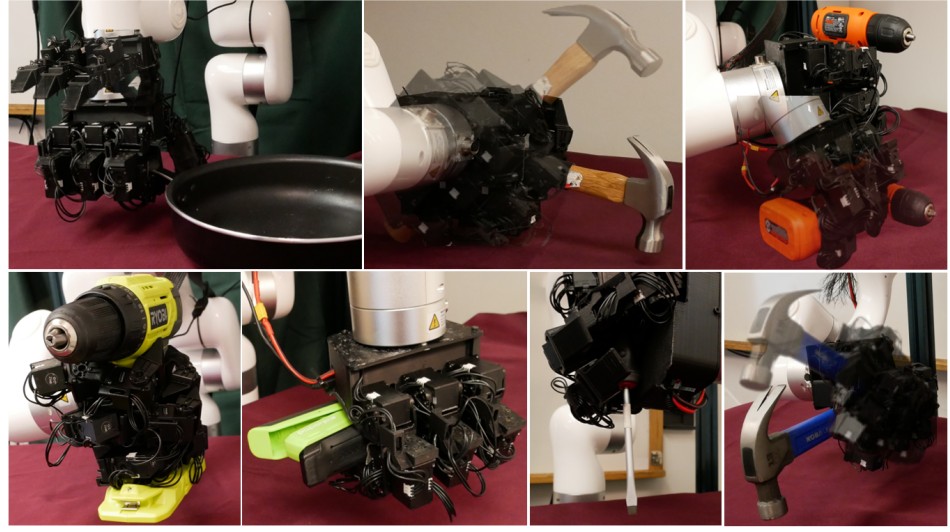

Figure 1: We use a single policy trained in simulation to pickup and grasp objects like hammers, drills, saucepan, staplers and screwdriver in different positions and orientations. An affordance model based on matching DINOv2 features is used to localize the object and move above the relevant region of the object. A blind reactive policy then picks up the object and moves it inside the palm to a firm grasp so that post-grasp motions like drilling, hammering, etc can be executed. Videos at `https://dexfunc.github.io/`.

**Abstract:** While there have been significant strides in dexterous manipulation, most of it is limited to benchmark tasks like in-hand reorientation which are of limited utility in the real world. The main benefit of dexterous hands over two-fingered ones is their ability to pickup tools and other objects (including thin ones) and grasp them firmly in order to apply force. However, this task requires both a complex understanding of functional affordances as well as precise low-level control. While prior work obtains affordances from human data this approach doesn't scale to low-level control. Similarly, simulation training cannot give the robot an understanding of real-world semantics. In this paper, we aim to combine the best of both worlds to accomplish functional grasping for in-the-wild objects. We use a modular approach. First, affordances are obtained by matching corresponding regions of different objects and then a low-level policy trained in sim is run to grasp it. We propose a novel application of eigengrasps to reduce the search space of RL using a small amount of human data and find that it leads to more stable and physically realistic motion. We find that eigengrasp action space beats baselines in simulation and outperforms hardcoded grasping in real and matches or outperforms a trained human teleoperator. Videos at `https://dexfunc.github.io/`.

**Keywords:** Functional Grasping, Tool Manipulation, Sim2real

## 1 Introduction

The human hand has played a pivotal role in the development of intelligence – dexterity enabled humans to develop and use tools which in turn necessitated the development of cognitive intelligence.

7th Conference on Robot Learning (CoRL 2023), Atlanta, USA.

[1, 2, 3, 4, 5] Dexterous manipulation is central to the day-to-day activities performed by humans ranging from tasks like writing, typing, lifting, eating, or tool use to perform end tasks. In contrast, the majority of robot learning research still relies on using two-fingered grippers (usually parallel jaws) or suction cups which makes them restricted in terms of the kind of objects that can be grasped and how they can be grasped. For instance, grasping a hammer using a parallel jaw gripper is not only challenging but also inherently unstable due to the center of mass of the hammer being close to the head, which makes it impossible to use it for the hammering function it is intended for. Although there are lots of recent works in learning control of dexterous hands, they are either limited to simple grasping or the tasks of in-hand reorientation [6, 7, 8, 9, 10, 11] which ignore the functional aspect of picking the object for tool use.

This paper investigates the problem of functional grasping of such complex daily life objects using a low-cost dexterous multi-fingered hand. For instance, consider the sequence of events that take place when one uses a hammer. First, the hammer must be detected and localized in the environment. Next, one must position their hand in a suitable pose perpendicular to the handle such that a suitable grasp pose may be initiated. A hammer may be feasibly grasped from both the hammer or the head and choosing the correct pose (also known as *pre-grasp pose*) requires an understanding of how hammers work. Next, the actual grasping motion is executed which is a high-dimensional closed-loop operation involving first picking up the hammer from the table and then moving it with respect to the hand into a firm power grasp. Power grasp is essential to ensure the stability of the hammer during usage. Once this is done, the arm can then execute the hammering motion while the hand holds it stably (*post-grasp trajectory*). Notably, the act of functional grasping, which is almost a muscle memory for humans, is not just a control problem but lies at the intersection of perception, reasoning, and control. How to do it seamlessly in a robot is the focus of our work.

Inspired by the above example, we approach the problem of functional grasping in three stages: predicting pre-grasp, learning low-level control of grasping, post-grasp trajectory. Out of these stages, visual reasoning is the critical piece of the first and third stage, while the second stage can be performed blind using proprioception as long as the pre-grasp pose is reasonable. To obtain the pre-grasp pose, we use a one-shot affordance model that gives pre-grasp keypoints for different objects in different orientations by finding correspondences across objects. To obtain these correspondences, we leverage a pretrained DinoV2 model [12] which is trained using self-supervised learning on internet images. This allows us to generalize across object instances. However, a more challenging problem is how to learn the low-level control for functional grasping the task itself.

We take a sim2real approach for the grasping motion in our approach. Prior approaches to sim2real have shown remarkable success for in-hand reorientation [7, 6] and locomotion [13, 14, 15, 16]. However, we observe that directly applying prior sim2real methods that have shown success in locomotion or reorientation yields unrealistic finger-gaiting results in simulation that are not transferrable to the real world. This is because grasping tools typically involve continuous surface contacts and high forces while maintaining the grasping pose – challenges which pose a significant sim2real gap and are nontrivial to engineer reward for. We introduce an action compression scheme to leverage a small amount of human demo data to reduce the action space of the hand from 16 to 9 and constrain it to output physically realistic poses. We evaluate our approach across 7 complex tasks in both the real world and simulation and find that our approach is able to make significant progress towards this major challenge of dexterous functional grasping as illustrated in Figure 1.

## 2 Method: Dexterous Functional Grasping

In this paper, we aim to combine the best of both human data and large scale simulation training to accomplish dexterous functional grasping in the real world. Given an object to grasp we use an affordance model to predict a plausible *functional* grasp pose for the hand. Then, we train a blind pickup policy to pickup the object and then grasp it tightly so that the arm may execute the post-grasp trajectory. Our method is divided into three phases - the pre-grasp, grasp and post-grasp (see Fig. 2)

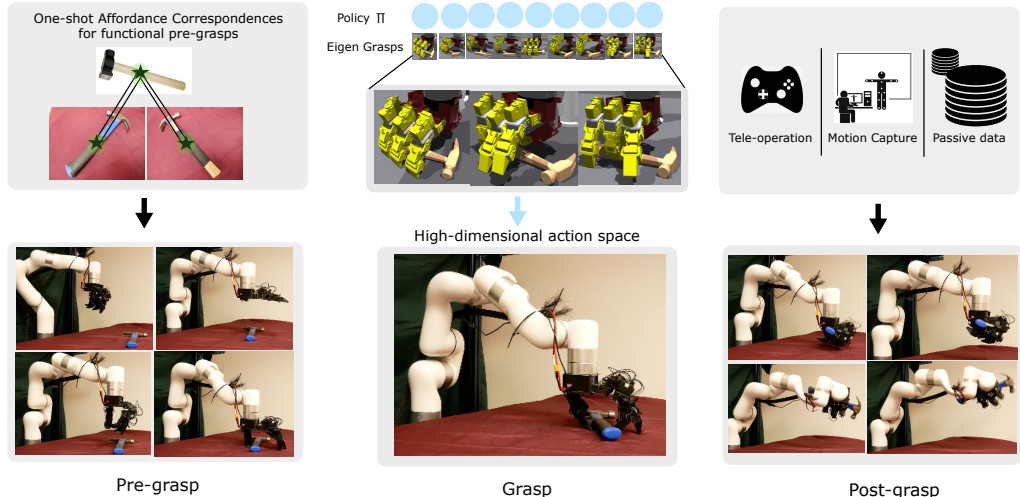

Figure 2: To get the pre-grasp pose we use a one-shot affordance model. After annotating one object we are able to get affordances for other objects in that category via feature matching. Given a new object, the arm is moved to that point and oriented perpendicular to the principal component of the object mask. The sim2real pickup policy is then executed and moves the object into a power grasp. After this, a post grasp trajectory can be safely executed.

In the pre-grasp phase, an affordance model outputs a region of interest of the object and we use the local object geometry around that region to compute a reasonable pre-grasp pose. We train a sim2real policy to execute robust grasps for pickup. However, in contrast to two fingered manipulation or locomotion where simple reward functions suffice, in the complex high-dimensional dexterous case it is easy to fall into local minima or execute poses in simulation that are not realizable in the real world. We therefore use human data to extract a lower-dimensional subspace of the full action space and run RL inside the restricted action space. Empirically, this leads to physically plausible poses that can transfer to real and stabler RL training.

## 2.1 Pre-grasp pose from affordances

An affordance describes a region of interest on the object that is relevant for the purpose of using it. This usually cannot be inferred from object geometry alone and depends upon the intended proper use of the object. For instance, by just looking at the geometry or by computing grasp metrics we could conclude that grabbing a hammer from the head or handle are both equally valid ways of using it. However, because we have seen other people use it we know that the correct usage is to grab the handle. This problem has been studied in the literature and one approach is to use human data in the form of videos, demos to obtain annotations for affordances. However, these are either not scalable or too noisy to enable zero-shot dexterous grasping.

Another approach is to leverage the fact that affordances across objects usually correspond. For all hammers, no matter the type the hammering, affordance will always be associated with the handle. This implies that feature correspondence can be used in a one-shot fashion to obtain affordances. In particular, we use Hadjivelichkov et al. [17], where for each object category we annotate one image from the internet with its affordance mask. To obtain the affordance mask for a new object instance we simply match DINO-ViT features to find the region which matches the specified mask. Since the mask may bleed across the object boundary we take its intersection with the segment obtained using DETIC [18]. Taking the center of the resulting mask gives us the keypoint $(x_{img}, y_{img})$ in image space corresponding to the pre-grasp position. To get the $z_{img}$, we project to the points $(x_{img}, y_{img})$ into the aligned depth image and then transform by camera intrinsics and extrinsics to get the corresponding point in the coordinate frame of the robot $(x_{robot}, y_{robot}, z_{robot})$. To get the correct hand orientation $\mathbf{q}$ we use the object mask obtained from DETIC and take the angle perpendicular to its largest principal component. Since there are three cameras, one each along $x, y, z$ axes (Fig. 7) we repeat this process

for each camera and pick the angle that has the highest affordance matching score (see Fig. **??**). This allows us to grasp objects in any direction, like upright drills and glasses.

Given the pregrasp pose, we first move the hand to a point at fixed offset $(x_{\text{robot}}, y_{\text{robot}}, z_{\text{robot}}) + \delta \mathbf{v}$ where $\delta \mathbf{v}$ is a fixed offset along the chosen grasp axis. We then move the finger joints to a pre-grasp pose with the joint positions midway between their joint limits. We found that same pre-grasp pose to work well across objects since our policy learns to adapt to the inaccuracies in the pre-grasp.

## 2.2 Sim2real for dexterous grasping

Once the robot is in a plausible pre-grasp pose it must execute the grasp action which involves using the fingers to grip the object and then moving it into a stable grasp pose. This requires high frequency closed-loop control. Further, this is typically a locally reactive behavior which can be accomplished using proprioception alone. Indeed, once we move our hand close to the object we wish to grasp we can usually pick it up even if we close our eyes. However, the challenge is that learning high frequency closed loop behavior typically requires a lot of interaction data which is missing from human videos and infeasible to scale via demos. In the past, sim2real has had remarkable successes in locomotion and in-hand dexterous manipulation in learning robust and reactive policies and we propose to use this method here.

Dexterous manipulation however presents a unique challenge because of its high-dimensional nature. It is easy for the hand to enter physically inconsistent poses or experience self collisions. Further, RL in high dimensional action spaces is unstable or sample efficient. We propose to leverage a small amount of human data to restrict the action space to physically realistic poses.

**Eigengrasp action space** A small number of human demos are often used to guide RL towards reasonable solutions like offline RL [19], DAPG [20]. However, the main problem with these is that they fail to learn optimal behavior from highly suboptimal demos. Further, the coverage of the demo data may be very poor which can artificially restrict the exploration space of the RL algorithm. We propose a simple alternative to these approaches which works from a few demos and can discover optimal behaviors even from suboptimal data. Our insight is that we have a very weak constraint on the behavior of the RL policy. We only care that the individual hand poses are realistic and not so much about the exact sequence in which they occur. We can therefore restrict the action space such that only realistic hand poses are possible.

In particular, suppose we are given a mocap dataset $\mathcal{D} = \{\tau_1, \ldots, \tau_n\}$ where $\tau_i = (\mathbf{x}_1, \ldots, \mathbf{x}_k)$ and $\mathbf{x}_i \in \mathbb{R}^{16}$ is a set of joints angles of the 16 dof hand. We perform PCA on the set of all hand poses to get 9 eigenvectors $\mathbf{e}_1, \ldots, \mathbf{e}_m$ where $m = 9$. These vectors are called eigengrasps [21] and have been classically used in grasp synthesis approaches. Here, we instead use it as a compressed action space for RL. Our policy predicts $m$-dimensional actions $\pi(\mathbf{o}_t) = \mathbf{a}_t \in \mathbb{R}^m$. The raw joint angles are then computed as a linear combination of eigenvectors $(\mathbf{a_t})_1 \mathbf{e}_1 + \ldots + (\mathbf{a_t})_k \mathbf{e}_k$. This transformation reduces the action dimension of the RL problem and decreases sample complexity in addition to enforcing realism. It also exploits the property that the convex combination of any two realistic hand poses is also likely to be realistic. Thus, doing PCA (as opposed to training a generative model) allows the policy to output hand poses that were not seen in the dataset. Empirically, we find that this stabilizes training and minimizes variation between different random seeds.

**Rewards** We train our policy to lift objects off the ground and them firmly grasp them in their hand. We find that a simple reward function that is a combination of two terms $r_{\text{threshold}}$ and $r_{\text{hand-obj}}$ is enough. The first, is a binary signal incentivizing the policy to pickup the object $r_{\text{threshold}}(t) = \mathbb{I}\left[(\mathbf{r}_{\text{obj}}(t))_z \geq 0.04\text{cm}\right]$ and the second is a sum of exponentials and an L2 distance to incentivize the object to be close to the palm of the hand

$$r_{\text{hand-obj}}(t) = \sum_{i=1}^{3} \exp\left(-\frac{\|\mathbf{r}_{\text{obj}} - \mathbf{r}_{\text{hand}}\|}{d_i}\right) - 4\|\mathbf{r}_{\text{obj}} - \mathbf{r}_{\text{hand}}\|$$

where $d_1 = 10$cm, $d_2 = 5$cm and $d_3 = 1$cm. The overall reward function is $r(t) = r_{\text{hand-obj}}(t) + 0.1 \cdot r_{\text{threshold}}(t) + 1$. Due to the eigengrasp parameterization we do not need any additional reward shaping terms.

**Policy Architecture**  We use a recurrent policy as that maps observations $\mathbf{o}_t \in \mathbb{R}^{16}$ to actions $\mathbf{a}_t \in \mathbb{R}^9$. A stateful policy is able to adapt to changes in environment dynamics better than a feedforward one. This allows our robot to adapt to slight errors in the pre-grasp pose from the affordance model. The policy observes the 7 dimensional target pose (position, quaternion) of the end-effector and the 16 joint angle positions of the hand.

**Training environment**  We want our policy to be robust to different surface properties and geometries and grasp them firmly. We therefore domain randomize the physical properties of the object, robot and simulation environment. We procedurally generate a set of hammers in simulation with randomized physical parameters. The hand is initialized in a rough pre-grasp pose with hand joint angles zeroed out. This corresponds to a neutral relaxed pose for the hand. The end-effector pose is initialized to be close to the real world pose obtained from the affordance model. The arm is kept close to the ground for 1s to allow the grasp to execute and then spun around in a circle. Episodes are terminated if the hand object distance exceeds 20cm. This spinning motion produces tight grasps and we see emergent behavior where the hand adjusts its grasp in response to changes in orientation. We also randomize physical properties of the simulation and add gaussian noise to observations and actions to simulate actuator noise (see Tab. 4).

## 2.3  Post-grasp trajectory

Once the object or tool is firmly grasped, since it is mounted on a 6-dof arm it can be moved arbitrarily in space to accomplish tasks such as screwing, hammering, drilling, etc. During training and evaluation we use either motion capture trajectories or define a set of keypoints and interpolate between them, but in principle these could be obtained from other sources such as internet video or third person imitation.

## 3  Experimental Setup

We demonstrate the performance of our method on a variety of objects, both similar and very dissimilar to the training objects like stapler, drill (light and heavy), saucepan, hammer (light and heavy). In our real world experiments, we aim to understand the reliability and efficiency of our method relative to an expert *teleop oracle* (20 hours) and a *hardcoded* grasping primitive. The former acts as an upper bound on the performance of the hardware while the latter is designed to show that large scale sim training yields a more robust policy than a grasping hardcoded.

In simulation, we test the effectiveness of our restricted action space and policy architecture. First, we compare against an *unconstrained* baseline that operates in the full 16 dimensional action space. Second, we compare against a policy that operates in the latent space of a *VAE* trained on the mocap dataset. Unlike our method, since a VAE is a generative model it can only output hand poses seen in the dataset and cannot extrapolate to new ones. Finally, we compare to a *feedforward* version of our method where the RNN policy is replaced by a feedforward one. This is designed to test whether recurrence helps in adaptation to domain randomization.

We experimentally validate the pre-grasp affordance matching [17] part of our pipeline separately. We compare against CLIPort [22] and CLIPSeg [23], two CLIP-based affordance prediction methods. CLIPort uses demonstration data to learn the correct affordances in a supervised fashion. CLIPSeg uses CLIP text and image features to zero-shot segment an object given a text prompt.

|  | Average Reward | | | Success Rate | | |
| --- | --- | --- | --- | --- | --- | --- |
|  | Hammer | Drill | Screwdriver | Hammer | Drill | Screwdriver |
| Unconstrained | $213.40 \pm 169.37$ | $102.12 \pm 36.12$ | $121.28 \pm 96.05$ | $0.60 \pm 0.55$ | $0.09 \pm 0.11$ | $0.46 \pm 0.45$ |
| VAE | $140.60 \pm 109.24$ | $83.34 \pm 43.32$ | $117.25 \pm 76.26$ | $0.30 \pm 0.44$ | $0.08 \pm 0.18$ | $0.25 \pm 0.41$ |
| Feed-forward | $232.80 \pm 175.59$ | $104.61 \pm 44.84$ | $153.19 \pm 105.83$ | $0.60 \pm 0.54$ | $0.21 \pm 0.19$ | $0.56 \pm 0.52$ |
| **Ours** | $\mathbf{327.40 \pm 11.61}$ | $\mathbf{129.03 \pm 22.58}$ | $\mathbf{211.13 \pm 11.14}$ | $\mathbf{1.00 \pm 0.00}$ | $\mathbf{0.23 \pm 0.16}$ | $\mathbf{0.95 \pm 0.10}$ |

Table 1: We measure the average reward and success rate of the trained policy in simulation. For each method we train a policy to hold the object close to the palm while arm spins. A success is counted when the arm does not drop the object at anytime. We see that our method outperforms the baselines and has significantly less variation between the runs. This is likely because the restricted action space makes the exploration problem easier and the physically plausible poses help keep the motion smooth. Each policy was trained randomized hammer but still generalizes to other different objects.

## 3.1 Hardware

We use the xarm6 with our own custom hand pictured in Fig. 7. The arm has 6 actuated joints, while the hand has 16 joints, four on each digit (three fingers and one thumb). An overhead calibrated D435 camera facing downward is used to obtain masks and affordance regions. The hand consists of Dynamixel servos mounted in a special kinematic structure designed to maximize dexterity [24]. We use an overhead D435 camera to obtain pre-grasp end-effector poses. Both the arm and the hand run at 30Hz. To teleoperate the hand and collect human demos for eigengrasps we use a Manus VR glove with SteamVR lighthouses which gives fingertip and hand positions which are then retargeted to our hand as in Figure 8.

## 3.2 Implementation Details

We use IsaacGym [25] as a simulator with IsaacGymEnvs for the environments and rl_games as the reinforcement learning library. The policy contains a layer-normed GRU with 256 as the hidden state followed by an MLP with hidden states 512, 256, 128. The policy is trained using PPO with backpropagation through time truncated at 32 timesteps. We run 8192 environments in parallel and train for 400 epochs.

# 4 Results and Analysis

## 4.1 Simulation Results

We train each baseline and our method for 400 epochs over 5 seeds. We find that ours beats all other methods primarily because it is stable with respect to the seed whereas the other baselines fluctuate widely in performance across seeds resulting in a high standard deviation and lower average overall performance. Note that our method also perfectly solves the training task for all seeds. This is likely due to a combination of two factors (a) the restricted action space nearly halves the action dimension (from 16 to 9), since the search space scales exponentially with action dimension this cuts down the space significantly and it is more likely that the algorithm discovers optimal behavior regardless of seed, and (b) since each hand pose is realistic and doesn't have self-collisions it leads to smoother and more predictable dynamics in simulation allowing the policy to learn better.

The RNN policy is also better and more stable than the feedforward variant as reported in Table 1. This is because (a) an RNN can use the hidden state to adapt to domain randomization (b) since the hand hardware does not output joint velocities, the feedforward policy has no idea of how fast the fingers are moving which can hinder performance. The RNN on the other hand is able to implicitly capture velocity of joints in the hidden state and this helps it to learn better.

## 4.2 Real World Results

We choose a variety of objects to compare against – hammer (light and heavy), saucepan, drill (light and heavy), stapler and screwdriver. Of these, hammer and saucepan are quite similar to the

|  | Hammer (unseen) | | Spatula (seen) | | Frying Pan (seen) | |
|---|---|---|---|---|---|---|
|  | Pick success | IoU | Pick success | IoU | Pick success | IoU |
| CLIPort | 2/10 | 0.034 | 6/10 | 0.15 | 7/10 | 0.15 |
| ClipSeg | 1/10 | 0.05 | 2/10 | 0.06 | 1/10 | 0.014 |
| Ours | **9/10** | **0.33** | **8/10** | **0.23** | **7/10** | **0.17** |

Table 3: We compare our affordance matching against CLIPort and CLIPSeg in terms of pick success rate and IoU between the predicted and ground truth affordance (human-annotated). We use the simulated CLIPort dataset for both unseen and seen objects. Our method outperforms CLIPort on both seen and unseen categories. CLIPSeg fails because it does not capture object parts such as the handle of the hammer.

training distribution because of the handle geometry while the drill, stapler and screwdriver have substantially different geometry. The heavy drill is especially challenging because of its narrow grip and unbalanced weight distribution. We run 10 trials per object per baseline in the real world (see Table 2). For all objects except the saucepan we execute a post-grasp trajectory where the object is picked up and waved around to test the strength of the grasp. For the saucepan we simply pick it up since waving it around is a safety hazard. During each trial, the orientation is randomized in the range $[-\pi, \pi]$ and position is randomized in the entire workspace $1\mathrm{m} \times 0.5\mathrm{m}$, the affordance model is run and the hand is moved to the pre-grasp pose. Videos at `https://dexfunc.github.io/`.

We obtain the hardcoded baseline by interpolating between the fully open and fully closed eigengrasp over 1s. This leads to the hand quickly snapping shut before the arm rises up. We find that this baseline performs poorly and gets zero success rate on many objects, especially thin ones. This is because in order to successfully grasp the object the thumb must retract closer to the palm. However, the timing of this is crucial, if the thumb retracts too early then the object flies back away from the hand. This is the most common failure case of this baseline that we observe. The hardcoded grasp succeeds for tall objects like an upright stapler or if the object happens to be in a favorable pose at the time of grasping.

|  | Success Rate ↑ | | |
|---|---|---|---|
|  | Teleop Oracle | Hardcoded | Ours |
| Hammer (heavy) | 0.5 | 0.0 | 0.8 |
| Hammer (light) | 0.6 | 0.3 | 0.9 |
| Sauce pan | 0.9 | 0.3 | 0.9 |
| Drill (heavy) | 0.9 | 0.2 | 0.5 |
| Drill (light) | 0.9 | 0.3 | 0.8 |
| Stapler | 0.9 | 0.3 | 1.0 |
| Screwdriver | 0.5 | 0.0 | 0.7 |

Table 2: We show functional grasping for a varied set of objects. We compare to a hardcoded pinch grasp and a trained teleoperator with a VR glove. The hardcoded baseline fails since the fingers push the object behind. Our method is able to beat the teleop oracle on challenging objects such as screwdriver, stapler and hammer.

The teleop oracle baseline was carried out with a Manus VR glove with the joints mapped one to one to the robot hand (ignoring the human pinky). This was teleoperated by a trained user ( 20 hours of experience). This was intended to serve as an upper bound of hardware capability. We find our method matches or slightly lags behind the oracle for drill (light) and saucepan. Surprisingly, for stapler, screwdriver and both hammers it even exceeds the oracle baseline. This is because these objects are heavy and sit close to the ground and require very swift and forceful motion which is also very precise in order to be successfully picked out. This is very hard to execute reliably for a human being, whereas our policy is able to do it well. We also find that our method is able to complete the task in a shorter time for the same reason.

### 4.3 Affordance Analysis

We experimentally validate the pre-grasp affordance matching part of our pipeline separately. We compare against CLIPort [22] and CLIPSeg [23] in terms of both pick success rate and IoU between the predicted and ground truth affordance (human-annotated). We run evaluation on the simulated CLIPort dataset for both unseen and seen objects (Table. 3). For our method, we annotate one exemplar from each category. To obtain affordance from CLIPSeg we prompt with the relevant part of the object such as "hammer handle". Note that Spatula and Frying Pan are present in the CLIPort training data while hammer is a new category.

Our method outperforms CLIPort on both seen and unseen categories. We observe that CLIPSeg fails to localize objects or is not able to capture the functional part of the object and only has understanding of the entire object as a whole (Fig. 6). While CLIPort is able to localize objects better but often predicts bounding boxes that are not functionally correct (such as the pan part of the sauce instead of the handle in Fig. 6).

## 5 Related Work

**In-hand dexterous manipulation:**   Dexterity in humans is the ability to manipulate objects within their hand's workspace [26, 27, 28]. Accordingly, in-hand reorientation has remained a standard, yet challenging task in robotics to imitate a human's dexterity. In recent years, there has been a surge of interest in this field and sim2real approaches have shown some success at reorienting objects [7, 29, 9, 30, 8] and also manipulating them [6, 31]. Other works bypass sim and directly learn in-hand manipulation through trial and error in the real world [11, 10]. Some other works use human demos to guide RL [20] and others directly use demos to learn policies [32].

**Dexterous grasping:**   While in-hand reorientation is an important task most of the uses of a dexterous hand involve grasping objects in different poses. Because of the large degrees of freedom, grasp synthesis is significantly more challenging. The classical approach is to use optimization [21, 33, 34]. This approach is still used today with the form or force closure objective [35, 36, 37]. Some methods use the contact between the object and the hand as a way to learn proper grasping [38, 39, 40, 41]. A VAE can be trained on these generated poses to learn a function that maps from object to grasp pose [36, 42]. Recent works leverage differentiable simulation to synthesize stable grasp poses [43]. Other works don't decouple this problem into a grasp synthesis phase and learn it end-to-end in simulation [44], from demonstrations [45, 46, 32] or teleoperation [47, 48].

**Functional Grasping:**   While simulation can be a powerful tool to optimize grasp metrics, functional affordances are usually human data since there may be more than one physically valid grasp pose but only one functionally valid one that allows one to use the object properly. Some approaches rely on clean annotations or motion capture datasets [49, 50, 51, 52] for hand object contact [53, 54, 55, 56, 57]. Some papers learn affordances from human images or video [58, 59] directly or through retargeting. These can however be noisy since they rely on hand pose detectors such as [60, 61] which are often noisy and difficult to learn from directly [45]. Some recent work in this area begin to target functional grasping using large scale datasets as a prior [62, 63, 64].

## 6 Limitations and Conclusion

We show that combining semantic information from models trained on internet data with the robustness of low-level control trained in simulation can yield functional grasps for a large range of objects. We show that using eigengrasps to restrict the action space of RL leads to policies that transfer better and are physically realistic. This leads to policies that are better to deploy in the real world on robot hand hardware.

The main failure case of our policy in the real world is due to incorrect pre-grasps from the affordance model. In particular, if the pre-grasp is such that the knuckle of the thumb joint lies over the object then the grasp fails since the hand cannot get the thumb around the object. One way to address this limitation is to equip the robot with local field of view around the wrist such that it can finetune its grasp even if the affordance model is incorrect.

Our method currently does not leverage joint pose information from the affordance model. While we found this to not be necessary in the set of objects we have, it might be useful in the case of more fine-grained manipulation such as picking up very thin objects like coins or credit cards.

## 7    Acknowledgements

We would like to thank Russell Mendonca, Shikhar Bahl and Murtaza Dalal for fruitful discussions. KS is supported by NSF Graduate Research Fellowship under Grant No. DGE2140739. This work is supported by ONR N00014-22-1-2096 and the DARPA Machine Common Sense grant.

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

# A    Grasping along multiple axes

In some cases, an object may be kept upright and a top-down angle of approach does not work. To deal with these cases, we setup three cameras along each axis (Fig. 7) and run affordance matching for each one. We finally pick the axis that has the highest score and move the hand along that axis to the pre-grasp pose. See Fig. 3, 4 for a vizualization. Empirically, we find that the confidence score is indeed always highest for the correct direction of approach.

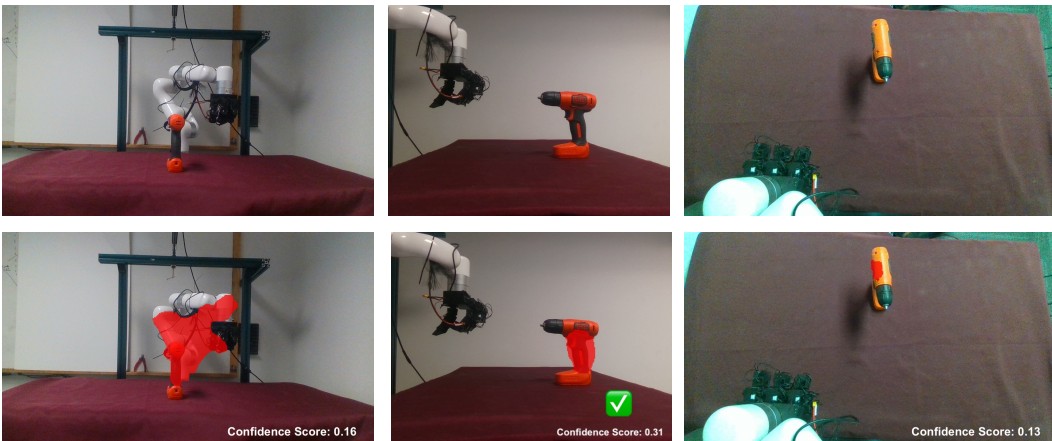

Figure 3: Affordance prediction for an upright drill from multiple angles. The best angle of approach is from the side and that is also the angle with highest affordance score. Our system picks this angle and executes a grasp.

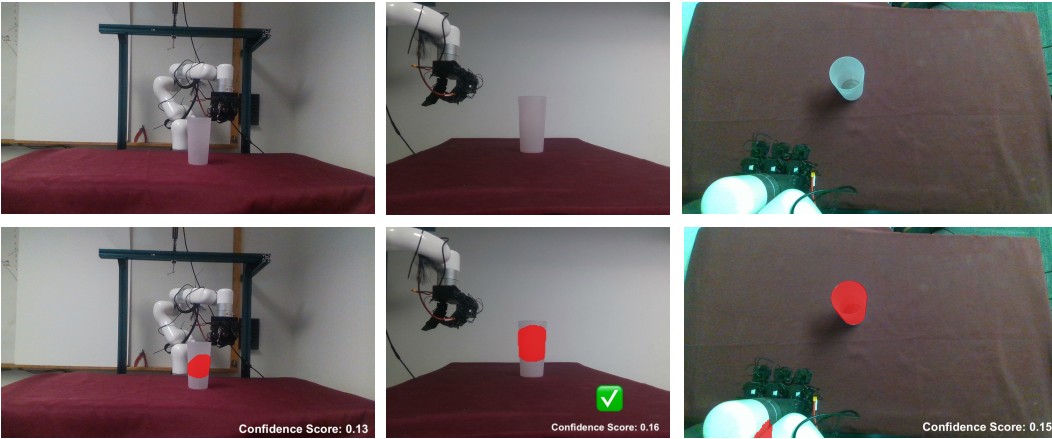

Figure 4: Affordance prediction for an upright mug from multiple angles. Our system picks the side angle with highest affordance score and executes a grasp.

## B    Training curves in simulation

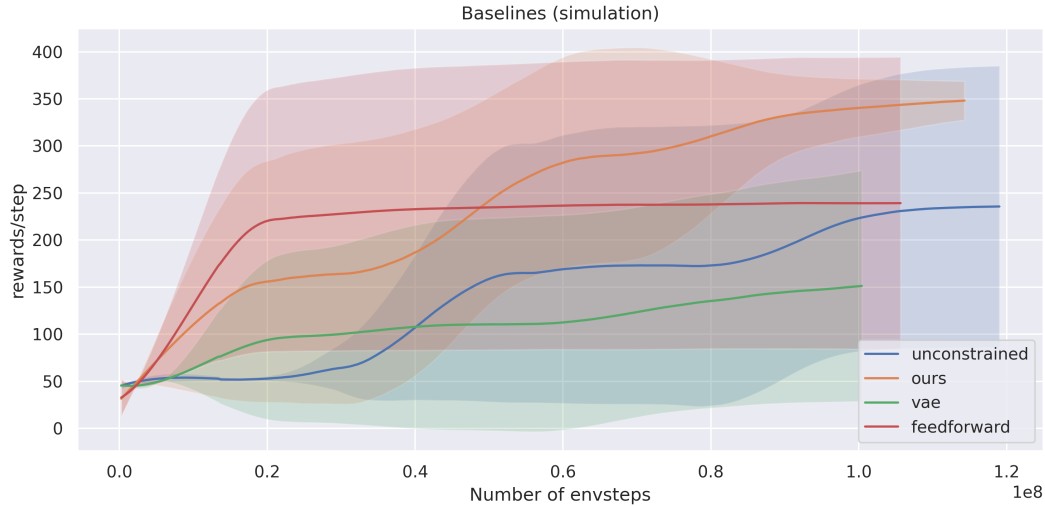

Figure 5: Training curves for baselines in simulation. Each baseline is run over 5 seeds. We see that ours outperforms the other baselines and also is more stable with respect to the seed. This is because of the lower dimensional action space.

## C    Qualitative results for affordance prediction

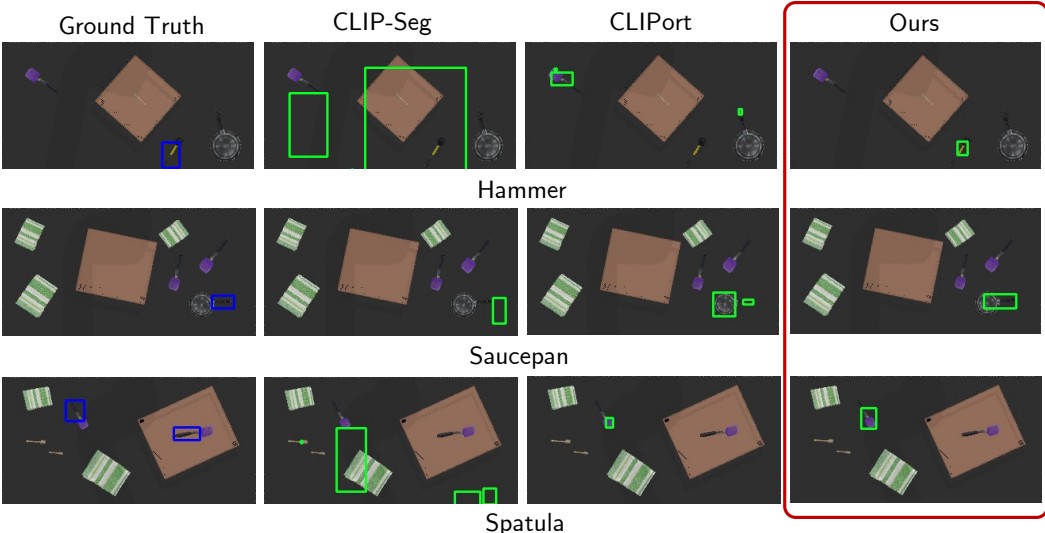

Figure 6: Qualitative comparisons of the affordance prediction from our method and CLIP-Seg, CLIPort. Overall, our method produces predictions that are more functionally aligned. CLIP-Seg is a zero-shot method and fails to localize the object correctly in many cases. CLIPort is able to localize the object but predicts grasp points that are not functional, for instance it predicts a bounding box around the head of the saucepan in addition to the handle.

# D   Hardware Setup

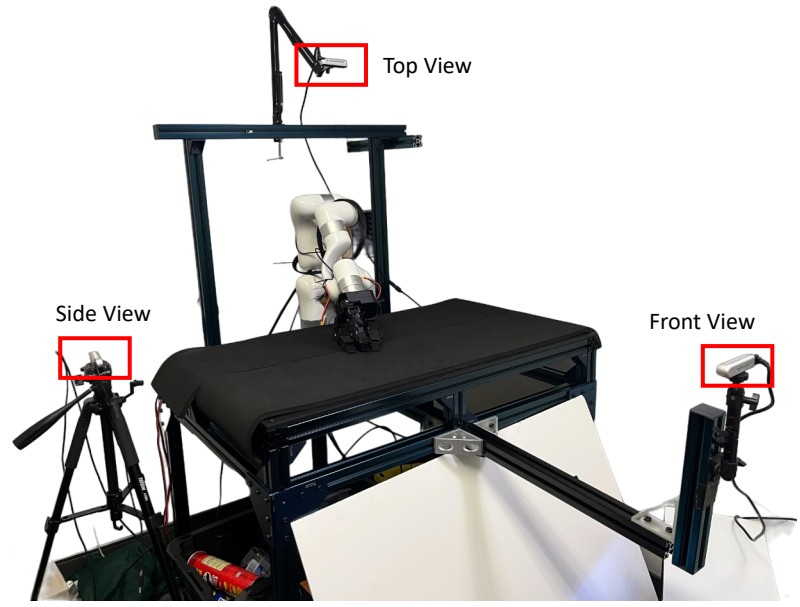

Figure 7: Hardware setup with LEAP hand mounted on xarm6 with one D435 along each axis.

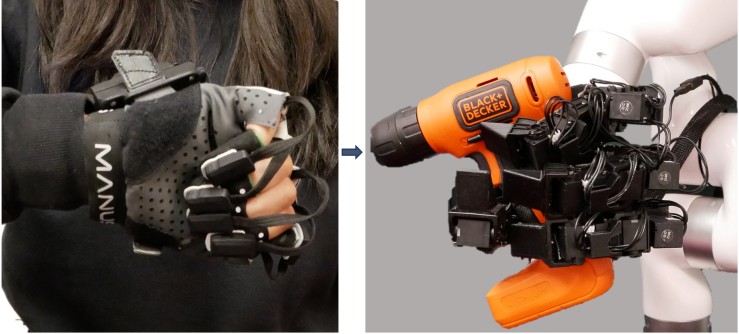

Figure 8: (left) the Manus VR glove we use to teleoperate our hand (right) the hand in the retargeted pose.

# E   Domain Randomization

For robustness, we domain randomize physics parameters as shown in Tab. 4.

| Name | Range |
|---|---|
| object scale | $[0.8, 1.2]$ |
| object mass scaling | $[0.5, 1.5]$ |
| Friction coefficient | $[0.7, 1.3]$ |
| stiffness scaling | $[0.75, 1.5]$ |
| damping scaling | $[0.3, 3.0]$ |

Table 4: domain randomization in simulation

