# OpenReview forum: "Dexterous Functional Grasping"
_robot-learning.org/CoRL/2023/Conference — CoRL 2023 Poster_

### Official Review · Reviewer_qb6f · 2023-07-19

**Confidence:** 4
**Originality:** Good
**Technical Quality:** Good
**Clarity Of Presentation:** Good
**Impact:** 3

**Recommendation:**

Weak Accept: I recommend accepting the paper, but will not argue for my recommendation if the majority of other reviewers have a different opinion.

**Review:**

The paper is well written and it is easy to follow. The authors motivate their problem properly with an example that helps the reader understand the complexity of the problem to be tackled.

One limitation of their introduction is the lack of proper introduction of previous approaches to the same problem. It is important to frame the proposed method in comparison with other works. This allows to frame the papers contribution in contrast with previous approaches. Is there any work aiming to solve dexterous grasping? How do they tackle their problem? What are their limitations? How does your proposed method improve over those limitations?

While the paper is not introducing novel methods in the algorithmic side, the proposed model might be interesting from a systems perspective. Nevertheless, the author fail to cite previous works on similar approaches in which the robot skill is represented as a two-phase problem of choosing a place to go and executing the skills. These methods are usually known as Coars-to-Fine approaches [1,2].

The paper presents a good evaluation of the grasp policy and compare their proposed EigenGrasp with other approaches that reduces the action space. Nevertheless, there is no any evaluation of the pre-grasp selection phase. Given, the pre-grasp selection is claimed as a contribution, it is mandatory to have an evaluation of that part.


**Strenghts**

1. The proposed problem is challenging and the proposed solution is interesting and makes sense.

2. The real-robot experiments for such a challenging task are highly appreciated.

**Weaknesses**

1. The lack of visual/tactile input in the RL policy limits the scope of the solution. The robot might need a visual signal to properly execute the grasping.

2. While the authors provide an interesting comparison of different approaches in the RL part, the paper is composed of two distinguished parts. The authos should provide a proper comparison of the affordance based approach with other affordance based methods, such as CLIP-Port [3] or Transported-Networks [4].

3. The paper is missing proper citations to Coarse-to-fine works that frame the problem in similar way.


[1] Coarse-to-Fine Imitation Learning: Robot Manipulation from a Single Demonstration, Johns

[2] Demonstrate Once, Imitate Immediately (DOME): Learning Visual Servoing for One-Shot Imitation Learning, Valassakis et al.

[3] Shridhar, Mohit, Lucas Manuelli, and Dieter Fox. "Cliport: What and where pathways for robotic manipulation." Conference on Robot Learning. PMLR, 2022.

[4] zeng, Andy, et al. "Transporter networks: Rearranging the visual world for robotic manipulation." Conference on Robot Learning. PMLR, 2021

**Quality Of The Limitations Section:**

Limitations are addressed clearly

**Questions For Rebuttal:**

The questions for rebuttal are in line with the highlighted weaknesses.

**Robotics Focus:**

Sufficient demonstration on hardware

**Summary Of Paper:**

The presented work deals with the problem of generating functional grasps for arbitrary objects  with dexterous robot hands. The paper proposes a two-phase method in which first an affordance point in the image is found and the robot is moved to a pre-grasp pose. Then, a learned RL policy is applied to properly grasp the object. To learn the RL policy, the authors propose a sim2real approach.

To obtain the pre-grasp pose in the image, the authors leverage DINO-ViT features. The authors propose matching the features of a previously annotated object with the object of interest. This feature matching will provide a (x,y) pixel in the image that is the one that best matches the features of the annotated image.

Once in the pregrasp pose, the grasping policy is learned by RL in a reduced dimensional action space that is linear to the original action space (Eigengrasps).

**Summary Of Recommendation:**

I suggest a rejection in the current stage of the work. Nevertheless, I believe that the paper could be an accepted paper if they properly address the current weaknesses.

---

> ### Author Response · Authors · 2023-08-12
> **Response to reviewer qb6f (with new experiments) [part 1/3]**
>
> Dear Reviewer,
>
> Thank you for the insightful feedback and for pointing out relevant related work! We are pleased to report that we were able to complete the experiments you suggested and added results on new objects as well - [https://dexfunc.github.io/rebuttal/additional.html](https://dexfunc.github.io/rebuttal/additional.html).
>
> We hope that these answers (+new experiments) address your concerns, and if so, we request the reviewer to consider updating the score. Otherwise, please let us know any further concerns that remain.
>
> ### Evaluation of affordance approach and comparison with CLIPort
> >*While the authors provide an interesting comparison of different approaches in the RL part, the paper is composed of two distinguished parts. The authors should provide a proper comparison of the affordance based approach with other affordance based methods, such as CLIP-Port [3] or Transported-Networks [4].*
>
> We experimentally validate the pre-grasp affordance matching [15] part of our pipeline separately. We compare against CLIPort and CLIPSeg in terms of both pick success rate and IoU between the predicted and ground truth affordance (human-annotated). We run evaluation on the publicly available simulated CLIPort dataset for both unseen and seen objects. For our method, we annotate one exemplar from each category. To obtain affordance from CLIPSeg we prompt with the relevant part of the object such as “hammer handle”. Note that Spatula and Frying Pan are present in the CLIPort training data while hammer is a new category.
>
> |         | Hammer (unseen) |       | Spatula (seen) |      | Frying Pan (seen) |       |  |
> |---------|-----------------|-------|----------------|------|-------------------|-------|--|
> |         | Pick success    | IOU   | Pick Success   | IOU  | Pick Success      | IOU   |  |
> | Cliport | 2 / 10          | 0.034 | 6/10           | 0.15 | 7 / 10            | 0.15  |  |
> | ClipSeg | 1/10            | 0.05  | 2/10           | 0.06 | 1/10              | 0.014 |  |
> | Ours    | **9/10**            | **0.33**  | **8/10**           | **0.23** | **7/10**              | **0.17**  |  |
>
>
> We see that our method outperforms CLIPort on both seen and unseen categories. We observe that CLIPSeg fails because it is not able to capture object parts such as the handle of the hammer and only has understanding of the entire object as a whole. We do not explicitly compare against TransporterNets since CLIPort authors find that their method outperforms it (Table 1 in the paper).
>
> For qualitative results see [https://dexfunc.github.io/rebuttal/affordance.html](https://dexfunc.github.io/rebuttal/affordance.html). We will add these results in the final version of the paper.

---

> > ### Author Response · Authors · 2023-08-12
> > **Response to reviewer qb6f [part 2/3]**
> >
> > ### Vision-based baseline
> > >*The lack of visual/tactile input in the RL policy limits the scope of the solution. The robot might need a visual signal to properly execute the grasping.*
> >
> > We would like to note that our overall method is not blind. The pre-grasp affordance matching phase critically relies on perception to localize the object and move to a functional pre-grasp pose. Once this pose is reached, we make the deliberate design choice to train a proprioception only grasping policy. This particular way of decomposing the problem is novel and has several benefits which we experimentally validate.
> >
> > As per your suggestion, we also train a vision-based RL policy based on existing work. Similar to [7] we train a policy with access to the 6D object pose and compare performance in simulation since the pose in the real world is not directly observable.
> >
> > |      			  | success rate|      |
> > |--------------------|----------------------|------|
> > |      			  | With Vision 	| Ours |
> > | Hammers (seen)  		  | 1.00 $\pm$ 0   | 1.00 $\pm$ 0 |
> > | Screwdrivers (unseen) |  0.80 $\pm$ 0.17     | 0.95 $\pm$ 0.10   |
> > | Drills (unseen) 		  | 0.23 $\pm$ 0.17   | 0.23 $\pm$ 0.16 |
> >
> > We see that on the seen objects the success rate is identical. This is because perception is naturally decoupled from the task of grasping. Once we have moved to the pre-grasp pose perception is no longer required. This is similar to how humans can grasp objects once they are close to the object even if it is dark and no vision is required. The object pose does not change much during the grasping process and object pose does not provide any additional information.
> >
> > We also note that our method performs better on out of distribution objects like screwdrivers. This is because the distribution shift in the proprioception-only observation space is smaller compared to when object pose is also added. The object pose for the screwdriver varies differently from that of the hammer and the vision-based policy encounters distribution shift and fails. We see that in the real world, our proprioception-only policy generalizes to diverse objects despite being trained only on rigid hammers.
> >
> > This decomposition also has an added benefit that it avoids the sim2real gap due to vision and tactile modalities which are typically hard to simulate. We will include this discussion and baseline in the paper.
> >
> > ### Comparison to prior work
> > >*One limitation of their introduction is the lack of proper introduction of previous approaches to the same problem. It is important to frame the proposed method in comparison with other works. This allows to frame the paper's contribution in contrast with previous approaches. Is there any work aiming to solve dexterous grasping? How do they tackle their problem? What are their limitations? How does your proposed method improve over those limitations?*
> >
> > The problem of functional grasping is challenging since it requires both knowledge of semantics and high frequency closed loop control. Closed loop control is required to execute the grasping action and move the object from the table to a stable grasp. This is typically learnt in simulation since it requires a lot of active data [7, 38, 50].
> >
> > However, semantics are required to know where to grasp an object from and cannot come from simulation data. For instance, a hammer must be grasped from the handle and not from the head. This fact can only be learnt from human data. Accordingly, some approaches learn affordances from passive data or annotations [32, 33, ] or collect demos and train using imitation learning [39-42]. However, human data in the joint space of the robot is limited and cannot be used to learn generalizable low-level control. Therefore, these approaches are restricted to simulation and affordance prediction alone or in the case of imitation learning, are limited in terms of diversity and generalization ability.
> >
> > In our work, we propose a novel pipeline that decomposes the problem to utilize the strengths of both simulation and passive data while avoiding their limitations. We show real-world results on a large diversity of objects. First, an affordance matching technique based on DiNO features [15] is used to predict where to grasp. Since this is trained on large quantities of passive data it is able to generalize to a large diversity of objects. Next, a proprioception-only grasping policy is trained on procedurally generated objects with domain randomization. Even though this policy is trained only on hammers it is able to generalize to a wide variety of objects in the real world. We will include this comparison to prior work in the introduction of our paper.

---

> > > ### Author Response · Authors · 2023-08-12
> > > **Response to reviewer qb6f [part 3/3]**
> > >
> > > ### Relationship to Coarse-to-Fine approaches
> > > >*The paper is missing proper citations to Coarse-to-fine works that frame the problem in similar way.*
> > >
> > > Thanks for pointing out these relevant papers. Coarse-to-fine works [1-4] indeed decompose the problem in a similar way to ours where the reaching and interaction policies are different. The robot learns to get close to the object in the same relative pose as the demonstration and then the demonstration is replayed to manipulate the object. Our approach differs in two key ways.
> > >
> > > - First, instead of relying on a demo for getting the bottleneck pose, we use DINO-ViT feature matching. DINO features can identify correspondences across object instances and even categories. Therefore, unlike coarse-to-fine methods our method is able to generalize to new instances and categories in a zero-shot manner.
> > >
> > > - Second, in coarse-to-fine the interaction policy is just the replayed demonstration. This means that if the object instance or category changes this policy will fail. In contrast, we leverage large-scale simulation pre-training to train reactive policies that generalize across a large range of objects.
> > >
> > > Thus, while coarse-to-fine approaches have a similar looking pipeline, they are very different in spirit. Coarse-to-fine approaches are designed to get maximal generalization from a single human demo, and this is restricted to generalization in the pose of the object. There is no generalization across instances or categories. On the other hand, our method is designed to leverage large amounts of passive and simulation data and use the strengths of each in the pre-grasp and interaction phases respectively. This is designed to allow generalization across instances as well as categories in the real world.
> > >
> > > We will add this discussion in the related work of our paper.
> > >
> > > ### Clarification about novelty
> > > >*While the paper is not introducing novel methods in the algorithmic side, the proposed model might be interesting from a systems perspective.*
> > >
> > > While we thank the reviewer for finding our proposed system interesting, we would like to note it is our novel conceptual and algorithmic contributions that make this possible:
> > >
> > > - **Conceptual Novelty**: Our first contribution is the specific pipeline consisting of pre-grasp, grasp and post-grasp we propose. As noted above, this allows us to generalize to a much greater variety of objects than in prior work. Our policy trained only on hammers is able to generalize to objects like deformable plushies and spray bottles. This allows us to leverage the best of both worlds, active simulation experience for grasping policy and internet passive data for pre-grasp and post-grasp. We believe this approach is more generally useful and can be coupled with other sim2real and affordance estimation techniques in the future.
> > > - **Technical Novelty**: Second, we propose a novel eigen-grasp action space reduction technique. While eigen-grasps have been used in past work for grasp optimization, to our knowledge ours is the first paper that proposes its use in RL as a structured action space. We show empirically that it leads to stable training and overall higher performance. We believe that this technique is generally useful for dexterous manipulation tasks, both for real-world and simulation training.

---

> > > > ### Author Response · Authors · 2023-08-12
> > > > **Response to reviewer qb6f (references)**
> > > >
> > > > [7] T. Chen, M. Tippur, S. Wu, V. Kumar, E. Adelson, and P. Agrawal. Visual dexterity: In-hand dexterous manipulation from depth.
> > > >
> > > > [15] D. Hadjivelichkov, S. Zwane, M. P. Deisenroth, L. de Agapito, and D. Kanoulas. One-shot326
> > > > transfer of affordance regions? affcorrs! In Conference on Robot Learning, 2022.
> > > >
> > > > [32] P. Grady, C. Tang, C. D. Twigg, M. Vo, S. Brahmbhatt, and C. C. Kemp. ContactOpt: Optimizing contact to improve grasps. In Conference on Computer Vision and Pattern Recognition (CVPR) 2021.
> > > >
> > > > [33] P. Mandikal and K. Grauman. Learning dexterous grasping with object-centric visual affordances. In 2021 IEEE International Conference on Robotics and Automation (ICRA)
> > > >
> > > > [34] S. Brahmbhatt, C. Ham, C. C. Kemp, and J. Hays. Contactdb: Analyzing and predicting grasp contact via thermal imaging. In Proceedings of the IEEE/CVF Conference on Computer Vision and Pattern Recognition (CVPR), June 2019
> > > >
> > > > [35] S. Brahmbhatt, A. Handa, J. Hays, and D. Fox. Contactgrasp: Functional multi-finger grasp synthesis from contact. In 2019 IEEE/RSJ International Conference on Intelligent Robots and Systems (IROS), pages 2386–2393, 2019
> > > >
> > > > [38] Y. Qin, B. Huang, Z.-H. Yin, H. Su, and X. Wang. Generalizable point cloud reinforcement learning for sim-to-real dexterous manipulation. In Deep Reinforcement Learning Workshop NeurIPS 2022.
> > > >
> > > > [39] K. Shaw, S. Bahl, and D. Pathak. VideoDex: Learning Dexterity from Internet Videos. In
> > > > Conference on Robot Learning (CoRL), 2022
> > > >
> > > > [40] Y. Qin, Y.-H. Wu, S. Liu, H. Jiang, R. Yang, Y. Fu, and X. Wang. Dexmv: Imitation learning for dexterous manipulation from human videos. In Computer Vision–ECCV 2022: 17th European Conference, Tel Aviv, Israel
> > > >
> > > > [41] A. Sivakumar, K. Shaw, and D. Pathak. Robotic telekinesis: Learning a robotic hand imitator by watching humans on youtube, Robotics Science and Systems (RSS) 2022.
> > > >
> > > > [42] A. Handa, K. Van Wyk, W. Yang, J. Liang, Y.-W. Chao, Q. Wan, S. Birchfield, N. Ratliff, and D. Fox. Dexpilot: Vision-based teleoperation of dexterous robotic hand-arm system. In IEEE International Conference on Robotics and Automation (ICRA)
> > > >
> > > > [50] S. Dasari, A. Gupta, and V. Kumar. Learning dexterous manipulation from exemplar object trajectories and pre-grasps. In IEEE International Conference on Robotics and Automation

---

> ### Author Response · Authors · 2023-08-15
> **Request for follow up response**
>
> Dear Reviewer,
>
> We gently wanted to bump this thread again as today is the last day of discussion.  Please let us know if you have any further questions.
>
> We hope to have addressed all the concerns you mentioned through additional experiments and ablations. You mentioned that the paper could be accepted if the current weaknesses were addressed, and if so, we would really appreciate your response and if you could consider updating the score.
>
> Thank you once again!
>
> -Authors

---

### Official Review · Reviewer_6xz6 · 2023-07-19

**Confidence:** 4
**Originality:** Very Good
**Technical Quality:** Very Good
**Clarity Of Presentation:** Excellent
**Impact:** 4

**Recommendation:**

Strong Accept: I recommend accepting the paper and will argue for my recommendation even if other reviewers hold a different opinion.

**Review:**

### Strengths
- The paper clearly formulates the research problem, describes a method to solve it, and evaluates it with experiments. The paper is well written and easy to understand.
- Strong real robot performance.
- Teleoperation as a baseline is a nice idea, because it also upper bounds the imitation learning baseline.

### Weaknesses
- The paper does not evaluate the accuracy of the affordance prediction method it uses.

**Quality Of The Limitations Section:**

Limitations are addressed clearly

**Questions For Rebuttal:**

Please include some information about evaluation of the affordance transfer method. For example, annotate the ground truth affordance on a couple of objects, place them in various poses and evaluate whether the proposed method is able to predict correct affordance. This is important because the downstream grasp step depends completely on affordance prediction. Do DINO-ViT features track correspondence across viewpoints? For example, if the prototypical Internet image used shows the drill lying down, but in the real world the drill is standing up resting on its handle.

**Robotics Focus:**

Sufficient demonstration on hardware

**Summary Of Paper:**

This paper proposes a system for functional grasping of household objects with a multifingered robot hand. The objects is placed without clutter on a tabletop and its image is matched to a canonical image of an object of the same class. Functional grasp affordance analysis on that image object is thus transferred to the real object, and the robot and is moved to the vicinity of the grasp location. Next, an RL policy trained in simulation is used to pick up and grasp the object. Finally, the arm is moved away using a keyframed motion.

Experiments are conducted on a real robot setup and the proposed algorithm is compared to a teleoperation policy that is recorded using cybergloves, and a hardcoded grasping primitive.

**Summary Of Recommendation:**

I am voting to "weak accept" this paper because of strong experimental performance on a real robotic system for a relevant and non-trivial problem - functional grasping with multi-fingered hand. The paper clearly mentions the research problem and shows that the proposed method addresses it.

### Post-rebuttal
The authors addressed both my concerns, so I am upgrading my rating to "strong accept".

---

> ### Author Response · Authors · 2023-08-12
> **Response to reviewer 6xz6 (with new experiments)**
>
> Dear Reviewer,
>
> Thank you for the insightful feedback and experiment suggestions!  We are pleased to report that we were able to complete the experiments you suggested and added results on more objects - [https://dexfunc.github.io/rebuttal/additional.html](https://dexfunc.github.io/rebuttal/additional.html).
>
> We hope that these answers (+new experiments) address your concerns, and if so, we request the reviewer to consider updating the score. Otherwise, please let us know any further concerns that remain.
>
> ### Accuracy of the affordance prediction module
> >*Please include some information about evaluation of the affordance transfer method. For example, annotate the ground truth affordance on a couple of objects, place them in various poses and evaluate whether the proposed method is able to predict correct affordance. This is important because the downstream grasp step depends completely on affordance prediction.*
>
> We experimentally validate the pre-grasp affordance matching [15] part of our pipeline separately. We compare against CLIPort and CLIPSeg in terms of both pick success rate and IoU between the predicted and ground truth affordance (human-annotated). We run evaluation on the publicly available simulated CLIPort dataset for both unseen and seen objects. We could not run the comparison on real data since that checkpoint of CLIPort is not available. For our method, we annotate one exemplar from each category. To obtain affordance from CLIPSeg we prompt with the relevant part of the object such as “hammer handle”. Note that Spatula and Frying Pan are present in the CLIPort training data while hammer is a new category.
>
>
> |         | Hammer (unseen) |       | Spatula (seen) |      | Frying Pan (seen) |       |  |
> |---------|-----------------|-------|----------------|------|-------------------|-------|--|
> |         | Pick success    | IOU   | Pick Success   | IOU  | Pick Success      | IOU   |  |
> | Cliport | 2 / 10          | 0.034 | 6/10           | 0.15 | 7 / 10            | 0.15  |  |
> | ClipSeg | 1/10            | 0.05  | 2/10           | 0.06 | 1/10              | 0.014 |  |
> | Ours    | **9/10**            | **0.33**  | **8/10**           | **0.23** | **7/10**              | **0.17**  |  |
>
>
> We see that our method outperforms CLIPort on both seen and unseen categories. We observe that CLIPSeg fails because it is not able to capture object parts such as the handle of the hammer and only has understanding of the entire object as a whole. We do not explicitly compare against TransporterNets since CLIPort authors find that CLIPort outperforms it (Table 1 in the paper).
>
> For qualitative results see [https://dexfunc.github.io/rebuttal/affordance.html](https://dexfunc.github.io/rebuttal/affordance.html). We will add these results in the final version of the paper.
>
> ### Tracking correspondence across viewpoints
> >*Do DINO-ViT features track correspondence across viewpoints? For example, if the prototypical Internet image used shows the drill lying down, but in the real world the drill is standing up resting on its handle.*
>
> Thank you for suggesting this example! It is true that with the setup in the submitted version of the paper which has a single top-down camera, DINO features would not be able to track correspondence in the case where the drill is standing up. However, there is a simple extension to the setup which allows for this.
>
> In addition to the camera pointing down along the z axis we add two more cameras along the x and y axis pointing towards the workspace. Next, to get the correct pre-grasp pose, we run affordance matching on each of the camera images and pick the one with the highest score. A rotation offset is applied along the axis of the chosen camera to get the correct orientation for the pregrasp. We are able to pick up upright objects like drills and mugs with this method. For result videos and a picture of the new setup see - [https://dexfunc.github.io/rebuttal/additional.html](https://dexfunc.github.io/rebuttal/additional.html). We will include this in the final version of the paper.

---

> ### Author Response · Authors · 2023-08-15
> **Request for follow up response**
>
> Dear Reviewer,
>
> We gently wanted to bump this thread again as today is the last day of discussion.  Please let us know if you have any further questions.
>
> We hope to have addressed all the concerns you mentioned through additional experiments and ablations, in particular the tracking of correspondences across viewpoints. If so, we would really appreciate your response and if you could consider updating the score.
>
> Thank you once again!
>
> -Authors

---

### Official Review · Reviewer_TMM4 · 2023-07-20

**Confidence:** 3
**Originality:** Good
**Technical Quality:** Good
**Clarity Of Presentation:** Good
**Impact:** 3

**Recommendation:**

Weak Accept: I recommend accepting the paper, but will not argue for my recommendation if the majority of other reviewers have a different opinion.

**Review:**

Strengths:
1. Several design choices are made to improve the performance of the grasping policy including using a recurrent neural network to encode the history of joint angles during the grasping motion and transforming the action space from the raw joint angles to linear weights for 9 eigengrasps computed from mocap data. These two strategies are shown to improve the success rate in simulation.
2. The problem this paper studies is relevant. Transitioning task-oriented grasping and tool use from parallel jaw grippers to dexterous hands is a logical and necessary step.

Weakness:
1. The paper claims to achieve functional grasping for objects in the wild. However, the diversity of the objects is rather limited. Most of the functional grasps seem to be on objects with large cylindrical handles. It would be necessary to show whether the proposed method can generalize to objects with more complex shapes such as scissors, cups, glasses, and knives.
2. The main algorithmic novelty seems to be limited to the use of the eigengrasps.
3. It’s unclear whether other vision-based policies can outperform the blind policy and whether visual observation is needed to generalize to more diverse objects. Comparisons with vision-based methods are needed to verify the benefit of the blind policy.
4. Even though this paper presents a complete pipeline including determining pre-grasp, picking up the object, and finally executing the manipulation trajectory. The first and last part of the pipeline is rather limited and lack experiment validation. There are many details that need to be considered to build a robust pipeline. For example, how to map pre-grasp poses to objects in random 6-DoF poses and how to recognize the poses of the object in hand after the grasping motion such that the post-grasp trajectory can be reliably executed.

**Quality Of The Limitations Section:**

Additional details required

**Questions For Rebuttal:**

1. Please discuss whether the method can generalize to a wider range of objects.
2. Please discuss and compare to existing vision-based policies.

**Robotics Focus:**

Sufficient demonstration on hardware

**Summary Of Paper:**

This paper addresses the functional grasping of objects with a dexterous hand. The process of dexterously manipulating an object is broken down into 3 stages including approaching the object with a pre-grasp that enables subsequent manipulation, stably picking up the object, and finally executing a post-grasp trajectory for tool use. This paper mainly investigates the second stage where a RL policy is learned to pick up objects in simulation and then transferred to the real world.

**Summary Of Recommendation:**

This paper mainly introduces a method to stably pick up objects with large handles with a dexterous hand. The policy is trained in RL and has shown reasonable transfer to the real world. This grasping module can potentially be combined with task-oriented pre-grasp and post-grasp adjustment to achieve dexterous tool use. However, only the grasping module is evaluated and the whole pipeline is left to be validated.

---

> ### Author Response · Authors · 2023-08-12
> **Response to reviewer TMM4 (with new results and experiments) [part 1/3]**
>
> Dear Reviewer,
>
> Thank you for the insightful feedback! We are pleased to report that as per your suggestion, we have included more objects in our real-world experiments [https://dexfunc.github.io/rebuttal/additional.html](https://dexfunc.github.io/rebuttal/additional.html).
>
> We hope that these answers (+new experiments) address your concerns, and if so, we request the reviewer to consider updating the score. Otherwise, please let us know any further concerns that remain.
>
> ### Generalization to different objects
> >*The paper claims to achieve functional grasping for objects in the wild. However, the diversity of the objects is rather limited. Most of the functional grasps seem to be on objects with large cylindrical handles. It would be necessary to show whether the proposed method can generalize to objects with more complex shapes such as scissors, cups, glasses, and knives.*
>
> We would like to highlight that our method is already able to handle a much greater variety of objects than prior work including many non-cylindrical ones ([link](https://dexfunc.github.io/#other_objects)) like stapler, deformable teddy bear and plushy doll and a spray bottle. In some cases, it exhibits emergent adaptive behavior where the grasp is adjusted on the fly based on proprioception ([link](https://dexfunc.github.io/index.html#emergent)). In contrast, prior work on dexterous grasping either uses teleoperation [41, 42], imitation learning [39, 40] or sim2real [38] and shows results on a much smaller set of objects. In our work, even though our method only sees hammers at training time, it can generalize to a much greater variety in the real world.
>
> Based on your suggestion, we show that our method generalizes to even more objects including a knife (with blade retracted for safety), and upright objects like a glass and drill. For the knife, we use the same policy as before. To grasp the drill and glass, we set up two additional cameras along the x and y axes and choose the orientation with the highest affordance matching score. We use the same grasping policy as before but with a rotation offset added to the arm. We will include these results in the paper. For the extended setup and videos see - [https://dexfunc.github.io/rebuttal/additional.html](https://dexfunc.github.io/rebuttal/additional.html#results)

---

> > ### Author Response · Authors · 2023-08-12
> > **Response to reviewer TMM4 [part 2/3]**
> >
> > ### Vision-based baselines
> > >*It’s unclear whether other vision-based policies can outperform the blind policy and whether visual observation is needed to generalize to more diverse objects. Comparisons with vision-based methods are needed to verify the benefit of the blind policy.*
> >
> > We would like to note that our overall method is not blind. The pre-grasp affordance matching phase critically relies on perception to localize the object and move to a functional pre-grasp pose. Once this pose is reached, we make the deliberate design choice to train a proprioception only grasping policy. This particular way of decomposing the problem is novel and has several benefits which we experimentally validate below.
> >
> > As per your suggestion, we also train a vision-based RL policy based on existing work. Similar to [7] we train a policy with access to the 6D object pose and compare performance in simulation since the pose in the real world is not directly observable.
> >
> > |      			  | success rate|      |
> > |--------------------|----------------------|------|
> > |      			  | With Vision 	| Ours |
> > | Hammers (seen)  		  | 1.00 $\pm$ 0   | 1.00 $\pm$ 0 |
> > | Screwdrivers (unseen) |  0.80 $\pm$ 0.17     | 0.95 $\pm$ 0.10   |
> > | Drills (unseen) 		  | 0.23 $\pm$ 0.17   | 0.23 $\pm$ 0.16 |
> >
> > We see that on the seen objects the success rate is identical. This is because perception is naturally decoupled from the task of grasping. Once we have moved to the pre-grasp pose perception is no longer required. This is similar to how humans can grasp objects once they are close to the object even if it is dark and no vision is required. The object pose does not change much during the grasping process and object pose does not provide any additional information.
> >
> > We also note that our method performs better on out of distribution objects like screwdrivers. This is because the distribution shift in the proprioception-only observation space is smaller compared to when object pose is also added. The object pose for the screwdriver varies differently from that of the hammer and the vision-based policy encounters distribution shift and fails. We see that in the real world, our proprioception-only policy generalizes to diverse objects despite being trained only on rigid hammers.
> >
> > This decomposition also has an added benefit that it avoids the sim2real gap due to vision and tactile modalities which are typically hard to simulate. We will include this discussion and baseline in the paper.
> >
> > ### Experimental validation of the pre-grasp step
> > >*Even though this paper presents a complete pipeline including determining pre-grasp, picking up the object, and finally executing the manipulation trajectory. The first and last part of the pipeline is rather limited and lack experiment validation. There are many details that need to be considered to build a robust pipeline. For example, how to map pre-grasp poses to objects in random 6-DoF poses and how to recognize the poses of the object in hand after the grasping motion such that the post-grasp trajectory can be reliably executed.*
> >
> > We experimentally validate the pre-grasp affordance matching [15] part of our pipeline separately. We compare against CLIPort and CLIPSeg in terms of both pick success rate and IoU between the predicted and ground truth affordance (human-annotated). We run evaluation on the publicly available simulated CLIPort dataset for both unseen and seen objects. We could not validate on real data since those checkpoints are not available for CLIPort. For our method, we annotate one exemplar from each category. To obtain affordance from CLIPSeg we prompt with the relevant part of the object such as “hammer handle”. Note that Spatula and Frying Pan are present in the CLIPort training data while hammer is a new category.
> >
> > |         | Hammer (unseen) |       | Spatula (seen) |      | Frying Pan (seen) |       |  |
> > |---------|-----------------|-------|----------------|------|-------------------|-------|--|
> > |         | Pick success    | IOU   | Pick Success   | IOU  | Pick Success      | IOU   |  |
> > | Cliport | 2 / 10          | 0.034 | 6/10           | 0.15 | 7 / 10            | 0.15  |  |
> > | ClipSeg | 1/10            | 0.05  | 2/10           | 0.06 | 1/10              | 0.014 |  |
> > | Ours    | **9/10**            | **0.33**  | **8/10**           | **0.23** | **7/10**              | **0.17**  |  |
> >
> > We see that our method outperforms CLIPort on both seen and unseen categories. CLIPSeg fails because it is not able to distinguish object parts such as the handle of the hammer and only has understanding of the entire object as a whole. For qualitative results see - [https://dexfunc.github.io/rebuttal/affordance.html](https://dexfunc.github.io/rebuttal/affordance.html). We will add these results in the final version of the paper.

---

> > > ### Author Response · Authors · 2023-08-12
> > > **Response to reviewer TMM4 [part 3/3]**
> > >
> > > ### Clarification on Novelty
> > > >*The main algorithmic novelty seems to be limited to the use of the eigen-grasps.*
> > >
> > > We would like to clarify that in addition to our **technical novelty** of parametrizing output actions via eigen-grasp, an important **conceptual novelty** of our paper is the specific decomposition of pre-grasp, grasping, and post-grasp that we propose. As noted above, this allows us to generalize to a much greater variety of objects than in prior work. Our policy trained only on hammers is able to generalize to objects like deformable plushies and spray bottles which are very different from the ones seen at training time.
> > >
> > > This decomposition also allows us to leverage different sources of data for different parts of the pipeline. Pre-grasp affordance does not require active experience and hence we can use passive internet data in the form of DINO-ViT features. Grasping does require active experience and we provide this with a fast simulator. Since grasping is reactive and very local in context, once a pre-grasp pose is achieved we can get away with domain randomization and a single object category (hammers) as opposed to simulating the set of all objects. In this way, the decomposition into pre-grasp, proprioception-only grasping and post-grasp allows us to leverage the best of both worlds, active simulation experience and internet passive data while avoiding their limitations. We believe this approach is more generally useful to the community and can be coupled with other sim2real and affordance estimation techniques in the future.
> > >
> > > [7] T. Chen, M. Tippur, S. Wu, V. Kumar, E. Adelson, and P. Agrawal. Visual dexterity: In-hand dexterous manipulation from depth. arXiv preprint arXiv:2211.11744, 2022.
> > >
> > > [15] D. Hadjivelichkov, S. Zwane, M. P. Deisenroth, L. de Agapito, and D. Kanoulas. One-shot326
> > > transfer of affordance regions? affcorrs! In Conference on Robot Learning, 2022.
> > >
> > > [38] Y. Qin, B. Huang, Z.-H. Yin, H. Su, and X. Wang. Generalizable point cloud reinforcement learning for sim-to-real dexterous manipulation. In Deep Reinforcement Learning Workshop
> > > NeurIPS 2022.
> > >
> > > [39] K. Shaw, S. Bahl, and D. Pathak. VideoDex: Learning Dexterity from Internet Videos. In387
> > > Conference on Robot Learning (CoRL), 2022
> > >
> > > [40] Y. Qin, Y.-H. Wu, S. Liu, H. Jiang, R. Yang, Y. Fu, and X. Wang. Dexmv: Imitation learning for dexterous manipulation from human videos. In Computer Vision–ECCV 2022: 17th European Conference, Tel Aviv, Israel
> > >
> > > [41] A. Sivakumar, K. Shaw, and D. Pathak. Robotic telekinesis: Learning a robotic hand imitator by watching humans on youtube, Robotics Science and Systems (RSS) 2022.
> > >
> > > [42] A. Handa, K. Van Wyk, W. Yang, J. Liang, Y.-W. Chao, Q. Wan, S. Birchfield, N. Ratliff, and D. Fox. Dexpilot: Vision-based teleoperation of dexterous robotic hand-arm system. In 2020396
> > > IEEE International Conference on Robotics and Automation (ICRA)

---

> ### Author Response · Authors · 2023-08-15
> **Request for follow up response**
>
> Dear Reviewer,
>
> We gently wanted to bump this thread again as today is the last day of discussion.  Please let us know if you have any further questions.
>
> We hope to have addressed all the concerns you mentioned through additional experiments and ablations, in particular the generalization to different objects and evaluation of the affordance pipeline. If so, we would really appreciate your response and if you could consider updating the score.
>
> Thank you once again!
>
> -Authors

---

### Official Review · Reviewer_Kobm · 2023-08-04

**Confidence:** 4
**Originality:** Very Good
**Technical Quality:** Good
**Clarity Of Presentation:** Good
**Impact:** 3

**Recommendation:**

Weak Accept: I recommend accepting the paper, but will not argue for my recommendation if the majority of other reviewers have a different opinion.

**Review:**

Overall, the clarity of this paper is very well presented, and clearly delivers an original approach to improving dexterous grasping using a combination of human demonstration and reinforcement learning, which they demonstrate on real hardware.

This paper shows a fairly well-developed set of comparisons across competing metods, including policies implemented as a VAE, feed-forward, and without the eigengrasp action space. It would be worth showing in the supplemental how different combinations of these other methods, for instance using the VAE with the unconstrained actions, could potentially improve the performance of either, since the VAE would implicitly learn a compressed representation of the actions, without relying on the eigengrasps as yet another lower-dimensional representation of the hand poses.

The experimental setup is also clearly described and demonstrated, although it is also very surprising that the tele-op "oracle" performs worse than the trained policy on lifting some of the objects, and it is certainly a strong positive result from this proposed method.

**Quality Of The Limitations Section:**

Additional details required

**Questions For Rebuttal:**

The main limitations I see from this work stem from the experimental setup and which methods are compared with in their approach. This is where some details on which features provide what sort of benefit to the overall system, and which ones are extraneous are in question. For instance, were there any concrete advantages to using DINO-ViT features when training the model, or would some form of CLIP embeddings with perform just as well? These are the kinds of comparisons and evaluations that could help Table 1 clearly demonstrate the key advantages to the approach taken. A mention of competing/similar features used for training similar kinds of systems would be useful to include in the related work and/or Limitation section. For example, see methods like [1-3].

Similarly, what are the limitations of existing works that attempt works in approaching dexterous grasping with a similar approach, such as Dasari et al. '23, Handa et al. '22, and others? These kinds of comparisons to what is presented can be simple without requiring additional evaluation, but the related works section currently does not reference these methods very clearly in the context of this work, besides the topical relevance. It would be more convincing if they are able to demonstrate the clear advantages that are posed by this method (which are there given the real world demonstrations that many of these past methods fail to show).

Lastly, It would be good for this work to highlight how this approach may benefit from larger scale datasets, and showing an additional experiment comparing the performance using fewer oracle demonstrations would be convincing in showing how this approach could be built on further by more data-collection.

[1] Wen, M., Kuba, J., Lin, R., Zhang, W., Wen, Y., Wang, J., & Yang, Y. (2022). Multi-agent reinforcement learning is a sequence modeling problem. Advances in Neural Information Processing Systems, 35, 16509-16521.
[2] Xie, Z., Lin, Z., Li, J., Li, S., & Ye, D. (2022). Pretraining in Deep Reinforcement Learning: A Survey. arXiv preprint arXiv:2211.03959.
[3] Turpin, D., Zhong, T., Zhang, S., Zhu, G., Liu, J., Singh, R., ... & Garg, A. (2023). Fast-Grasp'D: Dexterous Multi-finger Grasp Generation Through Differentiable Simulation. arXiv preprint arXiv:2306.08132.

**Robotics Focus:**

Sufficient demonstration on hardware

**Summary Of Paper:**

This work proposes a method for learning to grasp and execute functional grasping motions on objects with a dexterous hand by using a combination of learning from demonstrations and reinforcement learning. Their key contributions are a) using eigengrasps to reduce the action space of the hand from demonstrations and reduce the difficulty of exploration for reinforcement learning based methods, b) a sim to real approach that leverages a hierarchical policy which breaks the task into a pre-grasp, grasp, and post-grasp trajectory. Their method experimentally validates their contribution by demonstrating their method's superior success rate and performance in simulated and realistic robotic grasping and control environments.

**Summary Of Recommendation:**

I recommend this paper be accepted given it's strong demonstrated results on real hardware. It also showcases a novel use of eigengrasps as a way of simplifying the action space for dexterous grasping and manipulation. However, this work would benefit from clearly contextual comparisons to related work, which go largely unaddressed. Lastly, this paper includes a fairly convincing set of demonstrations, with results that seem to outperform the human demonstrator on specific objects, which should be built on further in future work in terms of scaling the approach.

---

> ### Author Response · Authors · 2023-08-12
> **Response to reviewer Kobm (with experiments) [part 1/2]**
>
> Dear reviewer,
>
> Thanks a lot for the insightful and positive review and experiment suggestions! We are pleased to report that we were able to complete the experiments you suggested and added results on new objects - [https://dexfunc.github.io/rebuttal/additional.html](https://dexfunc.github.io/rebuttal/additional.html).
>
> We hope that these answers (+new experiments) address your concerns, and if so, we request the reviewer to consider updating the score. Otherwise, please let us know any further concerns that remain.
>
> ### Comparison to CLIP-based features
> >*The main limitations I see from this work stem from the experimental setup and which methods are compared with in their approach. This is where some details on which features provide what sort of benefit to the overall system, and which ones are extraneous are in question. For instance, were there any concrete advantages to using DINO-ViT features when training the model, or would some form of CLIP embeddings with perform just as well? *
>
> We experimentally validate the pre-grasp affordance matching [15] part of our pipeline separately. We compare against two methods which utilize CLIP features – CLIPort and CLIPSeg. For metrics we use pick success rate and IoU between the predicted and ground truth affordance (human-annotated). We run evaluation on the publicly available simulated CLIPort dataset  for both unseen and seen objects. We could not evaluate on real world data because those checkpoints for CLIPort are not public. For our method, we annotate one exemplar from each category. To obtain affordance from CLIPSeg we prompt with the relevant part of the object such as “hammer handle”. Note that Spatula and Frying Pan are present in the CLIPort training data while hammer is a new category.
>
> |         | Hammer (unseen) |       | Spatula (seen) |      | Frying Pan (seen) |       |  |
> |---------|-----------------|-------|----------------|------|-------------------|-------|--|
> |         | Pick success    | IOU   | Pick Success   | IOU  | Pick Success      | IOU   |  |
> | Cliport | 2 / 10          | 0.034 | 6/10           | 0.15 | 7 / 10            | 0.15  |  |
> | ClipSeg | 1/10            | 0.05  | 2/10           | 0.06 | 1/10              | 0.014 |  |
> | Ours    | **9/10**            | **0.33**  | **8/10**           | **0.23** | **7/10**              | **0.17**  |  |
>
>
> We see that our method outperforms CLIPort on both seen and unseen categories. We observe that CLIPSeg fails because it is not able to capture object parts such as the handle of the hammer and only has understanding of the entire object as a whole. We do not explicitly compare against TransporterNets since CLIPort authors find that CLIPort outperforms it (Table 1 in the paper).
>
> For qualitative results see [https://dexfunc.github.io/rebuttal/affordance.html](https://dexfunc.github.io/rebuttal/affordance.html). We will add these results in the final version of the paper.

---

> > ### Author Response · Authors · 2023-08-12
> > **Response to reviewer Kobm [part 2/2]**
> >
> > ### Benefit from large-scale dataset
> > >*It would be good for this work to highlight how this approach may benefit from larger scale datasets*
> >
> > Thanks for this suggestion! Each part of our pipeline can benefit from a larger scale of dataset. First, the pre-grasp affordance will be more accurate. For instance, we observed a large improvement when we switched from DiNO v1 to v2 features. Second, the grasping policy can benefit from greater diversity of objects in simulation. This is an important direction for future work.
> >
> > >*...and showing an additional experiment comparing the performance using fewer oracle demonstrations would be convincing in showing how this approach could be built on further by more data-collection.*
> >
> > We would like to clarify that our method already requires minimal data collection. It does not require any demos and only requires the user to annotate a single prototype image with the correct affordance for each category. In total, to run evaluations for Table 2 in our paper we only had to annotate 5 images. We did not collect any demos.

---

> ### Author Response · Authors · 2023-08-15
> **Request for follow up response**
>
> Dear Reviewer,
>
> We gently wanted to bump this thread again as today is the last day of discussion.  Please let us know if you have any further questions.
>
> We hope to have addressed all the concerns you mentioned through additional experiments and ablations, in particular the comparison to CLIP based features for affordance prediction. If so, we would really appreciate your response and if you could consider updating the score.
>
> Thank you once again!
>
> -Authors

---

> > ### Comment · Reviewer_Kobm · 2023-08-16
> > **Follow-up to Rebuttal**
> >
> > Dear Authors,
> >
> > Thank you for adding this additional comparison in your results, it indeed does seem to highlight the key contribution of using DINO-ViT features while training the pre-grasp affordances, for both seen and unseen examples. After reviewing the updated table & evaluations you provided, it is assuring your method significantly outperforms the baseline's pick success rate with unseen hammer, and clear that the baseline using affordances performs better than CLIPSeg.
> >
> > Lastly, thank you for clarifying the number of annotated demonstrations needed to train the model.
> > I will update my review in the following period.

---

### Author Response · Authors · 2023-08-16
**Response to Area Chair**

Dear AC,

We thank the reviewers for the insightful feedback and highlighting positive aspects of our work: “strong real robot performance” (#6xz6), “real robot results on challenging tasks
 (#qb6f), “novel approach” (#Kobm), “well-developed comparisons” (#Kobm), “very well-presented” (#Kobm),  “well-written and easy to understand” (#6xz6, #qb6f).

We are pleased to report that we finished all the additional experiments / ablations mentioned by the reviewers and hope to have addressed all their concerns. Unfortunately, we haven’t received a reply from reviewers #TMM4, #qb6f, #6xz6 in time for us to respond further. We hope this is taken into consideration.

Summary of additional requested experiments/ablation:

- All the reviewers had questions about the accuracy affordance pipeline. We setup an evaluation pipeline for the affordance prediction and compared our method to CLIPseg and CLIPort (as per #Kobm’s suggestion) and found ours to be significantly better - [https://dexfunc.github.io/rebuttal/affordance.html](https://dexfunc.github.io/rebuttal/affordance.html)
- #TMM4 had questions about generalization to even more objects. We note that our method already generalizes to a larger diversity of objects than prior work. We also added new objects suggested by #TMM4 including knife and glass - [https://dexfunc.github.io/rebuttal/additional.html](https://dexfunc.github.io/rebuttal/additional.html)
- #TMM4 and #qb6f had questions about whether adding vision or tactile information would improve the RL grasping policy. As per their suggestion we followed prior work to train a baseline with prior work and found that our method does significantly better on out-of-distribution objects. We clarified why this is a direct consequence of our novel pipeline.
- Reviewers #qb6f, #TMM4 were concerned that our paper had limited novelty. We clarified that we have conceptual novelty in the form of the specific decomposition of pre-grasping, grasping and post-grasp that we propose. We also have algorithmic novelty in the use of parameterizing output actions via eigen-grasps.

---

### Decision · Program_Chairs · 2023-08-30

**Decision:**

Accept (Poster)

**Comment:**

The authors present a method to perform functional grasping of objects with a dexterous hand. They do this by breaking down the problem into multiple components. They perform affordance prediction from image observations, followed by a scripted move to a pre-grasp pose. They then use an RL policy in simulation to pick up and manipulate the object and transfer this to the real world. The authors present experiments on a real world system and break down evaluation of both the pre-grasp performance (added during rebuttal) and the post-grasp RL system. The experiments and ablations are quite comprehensive.

Reviewers highlighted that the rebuttal responses were compelling and this AC agrees that the rebuttal is particularly strong. Early feedback was primarily around experimental setup and missing ablations. In fact this was consistent with all reviewers. The authors addressed this feedback almost entirely by providing new experimental results and comparing (both experimentally and via updated manuscript content) with existing works..

While this did not impact the final recommendation, one AC comment would be to refrain from highlighting and summarizing reviewer comments that are not necessarily consistent with all reviewer feedback. As an example, “novel approach” is highlighted yet this is misleading as not all reviewers considered the system components novel, yet the author's response would suggest it is.